# Mental Health of Primary and Secondary School Teachers in the Remote Mountain Areas

**DOI:** 10.3390/medicina59050971

**Published:** 2023-05-17

**Authors:** Guoxiang Fang, Xiaofei Zhou, Yang Xin, Mei Li, Fang Li, Wenwen Zhang, Bo Li, Ying Wang

**Affiliations:** 1Department of Emergency, Xi’an No.3 Hospital, The Affiliated Hospital of Northwest University, Xi’an 710018, China; 2Department of Psychiatry, Xi’an International Medical Center Hospital, The Affiliated Hospital of Northwest University, Xi’an 710100, China; 3The College of Life Sciences and Medicine, Northwest University, No.229 Taibai North Road, Xi’an 710069, China

**Keywords:** mental health, remote area, primary and secondary school teachers, Symptom Checklist-90

## Abstract

*Background and objective*: Teaching is widely recognized as a highly stressful profession. Job stress leads to emotional exhaustion, which in turn triggers teacher attrition. The cost is estimated as USD 2.2 billion annually for teacher dropouts. It is therefore important to understand the mental state of teachers and the factors that may influence it in order to provide the appropriate early intervention. In the past, more attention has been paid to the mental status of teachers in economically developed cities, but less research has been conducted in remote cities. In this study, we selected primary and secondary school teachers in a typical area to assess their mental health, thereby contributing to the development of effective mental health education programs for teachers in primary and secondary schools. *Materials and methods:* In this study, 1102 teachers from a typical city in Ningxia Province, characterized by remote mountain areas, minority communities, and a low economic level, participated. The mental status of the teachers was assessed by a Symptom Checklist-90 (SCL-90). The effects of gender, age, level of education, place of work, and marital status on the total SCL-90 score were recorded and compared. The subscale scores of the SCL-90 and the differences among the respondents with various characteristics were analyzed. *Results*: In total, 1025 data were valid and used for statistical analysis. The effective rate of this study was 93.01%. The analysis showed that 25.17% of the subjects had possible mental problems. There were significant differences in age and marital status (*p* < 0.001). The score of teachers less than 30 years old was lower than that of other teachers (*p* = 0.001 vs. 30–39; *p* < 0.001 vs. 40–49; *p* < 0.001 vs. ≥50). The no-marriage teachers had the lowest score than the married group or others (*p* < 0.001 vs. married; *p* < 0.05 vs. others). Compared to the norm, teachers’ mental status was poor, especially in somatization (*p* < 0.001), obsessive-compulsive symptoms (*p* < 0.001), depression (*p* < 0.001), anxiety (*p* < 0.001), hostility (*p* < 0.001), phobic anxiety (*p* < 0.001), and psychosis (*p* < 0.001). There were gender differences in obsessive-compulsive symptoms (*p* < 0.05) and depression (*p* < 0.05). *Conclusions*: These data indicate that the mental status of these teachers is not optimistic, and married female teachers aged 40–55 years need to be given more attention. Mental health examinations can be incorporated into daily physical examination items to facilitate the timely detection and early intervention of negative emotions.

## 1. Introduction

Teaching, as an important part of the professional workforce, is recognized as a highly stressful profession [1,2], driven by the constant demands for attentional control and executive function. Teachers in primary and secondary schools perform important work in promoting the development of quality-oriented education [3]. Therefore, the mental and physical health of these teachers is of great concern to the whole of society [4,5,6]. Since 1978, China’s education system has been undergoing a series of reforms with the aim of promoting a shift in education from the traditional transmission of knowledge to quality education. In addition, more demands have been placed on teachers, especially primary and secondary school teachers. As a result, they face greater occupational stress [3,7]. With the development of Chinese society, the country’s demand for constructive talents is constantly increasing, and children in remote areas can only change their destiny through education, so the burden of imparting knowledge eventually falls on the shoulders of teachers. Chronic stress, if not effectively managed, overwhelms their coping capacity, causes burnout [8], disrupts the physiological systems, and ultimately leads to emotional exhaustion, decreased work efficiency, and reduced job satisfaction [9]. One study reports that stress and burnout contribute to teacher turnover [10] with approximately 40% of teachers leaving the profession after 5 years. The ILO/UNESCO Joint Committee of Experts on the Application of the Recommendations Concerning the Status of Teachers (1994) reported that accumulated stress contributes significantly to teacher turnover, with an estimated annual cost of USD 2.2 billion in teacher attrition, representing a significant downstream cost of teacher stress and burnout. Stress and burnout lead to physical and mental health problems, and the costs of treating these problems are also considerable [11,12].

Job stress leads to emotional exhaustion, which leads to teacher turnover [13,14]. There is growing evidence of a higher risk of mental disorders and work-related stress than in other professions [15,16]. Mental health problems are a common cause of absenteeism and a major reason for leaving the teaching profession [17]. Compared to other teachers, primary and secondary school teachers are exposed to greater job stress and are prone to more severe psychological problems [18]. There are some objective factors associated with this situation [19,20], such as outdated teaching conditions, high demands on teaching under the new educational reform, poor communication with parents, etc. On the other hand, the dramatic increase in the incidence of mental health problems among adolescents also puts more pressure on teachers. Some serious risks for students include bullying, truancy, and even suicidal behavior [21]. The teachers’ involvement in dealing with such problems also affects their mental health. School teachers are in a direct position to provide first aid to students with mental health problems.

The mental health of teachers is so important that it has attracted the attention of many researchers [22,23]. Interventions designed to reduce teachers’ stress and burnout are increasingly being implemented [24,25]. However, many of these interventions have been only marginal efforts due to relatively large individual differences [20,26]. More attention has been paid to the mental status of teachers in economically developed cities, but less research has been carried out in remote cities. In this study, 1102 teachers from a typical city in Ningxia Province, which is characterized by remote mountain areas, minority communities, and a low economic level, participated in the study. In addition, their mental status was assessed using the Symptom Checklist-90 (SCL-90). The effects of gender, age, education level, place of work, and marital status on the total SCL-90 score were recorded and compared. The subscale scores of the SCL-90 and the differences among the respondents with various characteristics were analyzed. In this study, we selected primary and secondary school teachers in a typical area to assess their mental health, thereby contributing to the development of effective mental health education programs for primary and secondary school teachers.

## 2. Subjects and Methods

### 2.1. Primary and Secondary School

In the education system of China, primary school or elementary school is a school in which children receive primary or elementary education from the ages of about 6 to 12 years, coming before secondary school. A middle school (also called a junior high school) is a school that children attend after primary school in places that use three levels of schooling, typically between the ages of about 13–15, which has no significant difference from the definition of secondary school.

### 2.2. Subject Participation

In total, 45 schools from a typical city in Ningxia Province, characterized by remote mountain areas, minority communities, and a low economic level, participated in the study. We made the subjects scan a QR code with their mobile phone to receive the questionnaire, then collected the data remotely and analyzed the results. In total, 1102 questionnaires were distributed and 1025 questionnaires were collected. The effective rate of this study was 93.01%. There were 401 males (39.8 ± 7.1 years) and 624 females (40.2 ± 9.1 years) in the study. Almost all (940, 91.70%) had a bachelor’s degree or higher. In total, 600 came from a primary school and the other 436 came from a secondary school. The age range of the participants was from 27 to 58 years. They were divided into four different subgroups according to age stage, namely, younger than 30 years (177, 17.27%), 30 to 39 years (312, 30.44%; including 30 years), 40 to 49 years (356, 34.73; including 40 years), and 50 years and older (180, 17.56%). The marital status of the participants was analyzed according to married, unmarried, and other, with 892 married, 97 unmarried, and 36 other. The total SCL-90 scores for married, unmarried, and other participants were 1.60 ± 0.58, 1.84 ± 0.72, and 1.36 ± 0.46, respectively (Table 1).

### 2.3. Psychological Test Tool: Symptom Checklist-90

The Symptom Checklist 90 (SCL-90), proposed by Derogatis, has been widely used worldwide [27]. It has shown good reliability and validity in discriminating psychosomatic symptoms [28,29]. According to the results of Derogatis and other studies, the validity coefficient of each symptom ranges from 0.77 to 0.90, indicating that the rating results of this scale have high authenticity and can better reflect the severity and change of the patient’s condition. The Chinese version of the SCL-90 has been translated by Wang [30]. The translated version of the scale includes ten symptom factors, i.e., somatization, obsessive-compulsive, interpersonal sensitivity, depression, anxiety, hostility, phobia, paranoia, and psychoticism, as well as eating and sleeping. Scores ranged from 1 to 5, and the higher the total score, the more severe the mental problem. A score of ≥2 points per factor indicates ‘mild symptoms’, and ≥3 points indicates ‘at least moderate symptoms’. The following conditions indicate a positive screening. A total score ≥160 points, the number of positive items ≥43 points, or the score per factor ≥2 points indicates a ‘positive condition’. For the translated version of the scale, Jin and Wu published a set of data of Chinese norms in 1986 that were based on data from 1388 patients [31]. Since then, many application studies have been carried out in China, and these studies mainly involving surveys of the general population and studies of barriers to large-scale screening.

### 2.4. Statistical Analyses

The SCL-90 total score and its ten subscale scores were of primary interest in this study. The SCL-90 data were compared with previously published norms. Statistical analyses were performed using SPSS 22.0 software (IBM-SPSS Inc., Chicago, IL, USA). A two-tailed significance test was used and the level of statistical significance was set at *p* < 0.05. The measurement data were expressed as mean ± standard deviation (x ± s), and the count data were expressed as a percentage (%). Differences in the SCL-90 scores among the respondents with various characteristics were analyzed using Student’s t-tests or one-way analysis of variance (ANOVA) tests and the Wilcoxon test was used to analyze the gender difference in the mental status of the subjects.

## 3. Results

### 3.1. Characteristics of Participants

The mean age of the 1025 teachers was 39.6 ± 7.25 years. More than half were female (60.9%) and most of the sample were married (87.0%) and had a university degree (91.7%). There were no significant differences in the SCL-90 scores according to gender, education, and workplace (gender: *p* = 0.304; education: *p* = 0.114; workplace: *p* = 0.849). Other demographic information is presented in Table 1. There were significant differences in the age and marital status (*p* < 0.001). Teachers aged less than 30 years had lower scores than other teachers (*p* = 0.001 vs. 30–39; *p* < 0.001 vs. 40–49; *p* < 0.001 vs. ≥50), and teachers aged 30–39 years had lower scores than teachers aged 40–49 years (*p* < 0.05 vs. 40–49; *p* < 0.05 vs. ≥50). There was no difference between the age of 40 and 49 and the ≥50 age groups (*p* > 0.05). The unmarried teachers had a lower score than the married group or others (*p* < 0.001 vs. married; *p* < 0.05 vs. others).

### 3.2. The Status of Mental Health

Of these 1025 individuals, 25.17% (258/1025) had possible mental health problems and 22.1% (57/258) of these individuals had moderate and severe mental health problems that might require further examination to clarify the presence of mental disorders. There was no gender difference in the mental status of the subjects (Z = −0.584, *p* = 0.559) (Table 2).

### 3.3. Subscales Score among Respondents

Compared to the norm, teachers’ mental status was poor, especially in somatization, obsessive-compulsive symptoms, depression, anxiety, hostility, fear anxiety, and psychosis. We made a comparison of the subscale scores between the respondents and the national norm. There were significant differences among respondents in somatization (t = 11.76, *p* < 0.001), obsessive-compulsive symptoms (t = 11.80, *p* < 0.001), depression (t = 7.38, *p* < 0.001), anxiety (t = 7.55, *p* < 0.001), hostility (t = 5.68, *p* < 0.001), phobic anxiety (t = 9.13, *p* < 0.001), and psychosis (t = 7.00, *p* < 0.001). However, when compared with the national norm, paranoid ideation showed no significant statistical difference (Table 3). We also compared the gender differences for each factor and found the differences in obsessive-compulsive symptoms (t = 2.285, *p* < 0.05) and depression (t = 2.105, *p* < 0.05). Compared to male teachers, female teachers might have slightly more pronounced phobic anxiety. However, there was no difference (t = 1.939, *p* = 0.053) (Table 4).

## 4. Discussion

The data showed that 25.17% of the subjects had possible mental health problems. Age and marital status were important influencing factors. Compared to the norm, the mental status of the teachers was poor, especially for somatization, obsessive-compulsive symptoms, depression, anxiety, hostility, fear anxiety, and psychosis, and there were gender differences for obsessive-compulsive symptoms and depression.

China’s education reform began in 1978, the year China launched its reform and opening-up policy, and education reform became an important part of it. Since then, Chinese education has undergone several reforms, including reforms in compulsory education, higher education, and vocational education. One of the most important reforms was the ‘New Curriculum Reform,’ which started in 1999 and aimed to promote the transformation of education from traditional knowledge transfer to quality education. Teaching can be a high-pressure profession. Teachers are responsible for educating students and leading them to success, which can be a challenging task. They must create lesson plans, grade assignments, manage classroom behavior, communicate with parents, and keep up with the latest teaching methods and technologies. In addition, teachers are often evaluated based on their students’ academic performance, which can add to the pressure. Numerous studies have shown a strong link between stress and the development of mental disorders. Chronic stress can cause or exacerbate psychiatric disorders, such as major depressive disorder [32,33,34]. Studies from suburban schools in Chile show that 28.6% of school teachers may have mental health problems and that teachers have a 2, 2.5, and 3 times higher risk of mental disorders than that of clerks, health care professionals, and blue-collar workers, respectively [35]. In our study, we found similar results. Compared to the standard level, the mental status of teachers was poor, which is consistent with previous studies [36,37,38]. Among these 1025 individuals, 25.17% have possible mental problems and 22.1% of these subjects have moderate and severe mental problems, which may require further examination to clarify the presence of mental disorders. In remote mountain areas of Shanxi Province, the mental status of 182 teachers was significantly lower than the national norm [36] and rural middle school teachers had a low level of mental health [37,38]. The mental health level of teachers has received considerable attention, and an increasing number of psychological interventions have been provided [35].

Teachers’ mental health can be affected by both external and internal factors. External factors include heavy workloads, feelings of injustice, low pay, lack of administrative support, unsatisfactory teaching environment and facilities, limited staffing, and lack of mental health training. These external factors can lead to stress, burnout, and may have exacerbated teachers’ mental health problems. Internal factors can include personal life stressors, such as financial or relationship problems, as well as individual personality traits and coping mechanisms. Teachers who are more resilient and have effective coping strategies may be better equipped to manage external stressors and maintain good mental health. In this study, we focused on the internal factors and found that age and marital status were important influencing factors. In combination with a previous study, the behavioral response to chronic stress showed sex-dependent changes—stress is anxiolytic in men and anxiogenic in women [39]. In our study, there were significant differences in age and marital status. The score of teachers under 30 years old was lower than that of other teachers. The unmarried teachers had a lower score than the married group or others. These two aspects are consistent, and young people up to the age of 30 tend to be unmarried people. In addition, young people have been in the workforce for a shorter period, so the relative pressure may be less. In our study, married female teachers aged 40–55 may have suffered the maximum possible pressure due to long working years, having heavy teaching loads, and having heavy family responsibilities.

As we all know, stress is the body’s psychophysiological response to unpredictable stressors [40,41]. Stress exposure not only affects the psychological state but also has a broad and significant impact on physiological function. Chronic stress is associated with maladaptive responses and produces detrimental effects on the body [40]. The teaching profession is characterized by an above-average rate of psychosomatic and mental health impairment due to work-related stress. In this study, we analyzed the total SCL-90 scores and found that there were significant differences in somatization, obsessive-compulsive symptoms, depression, anxiety, hostility, phobic anxiety, and psychoticism. The teachers had significant somatization symptoms. The possible reason may be related to the lack of positive solutions in the face of stress. Thus, psychological stress is relieved by triggering somatic symptoms. In the long term, this negative coping style can lead to psychosomatic disorders that affect the quality of teaching and life satisfaction. In addition, the teachers have more obvious obsessive-compulsive symptoms. It depends on the professional characteristics of teaching. Teaching is a very demanding profession, requiring exemplary behavior and academic rigor, which can easily lead to perfectionism and obsessive personality tendencies.

The high and persistent psychological distress among school teachers makes it particularly important to take an early preventive intervention, and the early identification of problems is particularly important in providing solutions for teachers who need and want support. Maintaining and improving teachers’ mental health is important because of its potential impact on students’ mood, achievement, and student–teacher relationships [42]. First, certain measures are needed to improve teachers’ mental health, including a better classroom environment, living conditions, educational policies, and social support systems. Second, society should strengthen education and help teachers to develop rational perceptions and realistic catharsis and to eliminate their psychological exhaustion [43]. In addition, local governments should provide professional training and career development plans for school teachers in remote areas to prevent their knowledge burnout.

## 5. Limitations of This Study

There are several limitations to our study. The first is the assessment tool. The data collected in this study were self-reported. This could lead to common methodological biases. Second, the participants may have concealed their true psychological status due to fear of leaking private information and the reputation of the school. Third, the Chinese norms of the SCL-90 were published in 1986, although they have their own validity and reliability, and are still in use today. However, if there are newly adapted Chinese norms of the SCL-90, we will use the new norms in our clinical trial. In addition, the possible affected stressors were not investigated. Finally, this study was conducted in a remote area of China, which may have biased the results.

## 6. Conclusions

This study shows that the mental status of primary and secondary school teachers in a remote mountain area is not optimistic. More than 1/4 may have mental health problems, and married female teachers aged 40–55 especially need to be paid more attention. Mental health examinations could be included in annual physical examinations to facilitate the timely detection and early intervention of negative emotions. It is important for schools and districts to provide support and resources to help teachers manage stress and maintain good mental health. This may include access to counseling services, professional development on stress management and self-care, and a positive school culture that values teacher well-being.

## Figures and Tables

**Table 1 medicina-59-00971-t001:** SCL-90 total scores of respondents by sociodemographic characteristics.

Characteristics	*n*	SCL-90 Score	t or F Value	*p*
Gender			−1.028	0.304
Male	401	1.56 ± 0.61		
Female	624	1.60 ± 0.56		
Education			1.582	0.114
University and above	940	1.58 ± 0.57		
Others	85	1.68 ± 0.71		
Workplace			−0.190	0.849
Primary school	600	1.58 ± 0.60		
Middle school	425	1.59 ± 0.55		
Age (years)			16.40	<0.001
Less than 30	177	1.35 ± 0.40		
Between 30 and 39	312	1.55 ± 0.50		
Between 40 and 49	356	1.68 ± 0.63		
More than 50	180	1.70 ± 0.67		
<30 vs. 30–39				0.001
<30 vs. 40–49				<0.001
<30 vs. ≥50				<0.001
30–39 vs. 40–49				0.022
30–39 vs. ≥50				0.028
40–49 vs. ≥50				0.975
Married			9.89	<0.001
Yes	892	1.60 ± 0.58		
No	97	1.84 ± 0.72		
Others	36	1.36 ± 0.46		
Yes vs. no				<0.001
Yes vs. others				0.131
No vs. others				0.001

Notes: All data are shown as the mean ± standard deviation. Married others: divorced/widow/widower. Abbreviation: SCL-90, Symptom Checklist-90.

**Table 2 medicina-59-00971-t002:** Gender difference in the severity of mental problems.

	Male	Female	Total	Z	*p*
None	303 (75.6%)	458 (73.4%)	761	−0.584	0.559
Mild	72 (18.0%)	135 (21.6%)	207		
Moderate	23 (5.7%)	28 (4.5%)	51		
Severe	3 (0.7%)	3 (0.5%)	6		
Total	401	624	1025		

**Table 3 medicina-59-00971-t003:** Comparison of scores on the SCL-90 of respondents with the national norm.

Subscales	Respondents (*n* = 1025)		National Norm (*n* = 1388)		t	*p*
	M	SD	M	SD		
Somatization	1.62	0.67	1.37	0.48	11.76	<0.001
Obsessive compulsive	1.87	0.69	1.62	0.58	11.80	<0.001
Interpersonal sensitivity	1.58	0.63	1.65	0.51	−3.45	0.003
Depression	1.65	0.67	1.50	0.59	7.38	<0.001
Anxiety	1.54	0.62	1.39	0.43	7.55	<0.001
Hostility	1.59	0.64	1.48	0.56	5.68	<0.001
Phobic anxiety	1.39	0.56	1.23	0.41	9.13	<0.001
Paranoid ideation	1.45	0.57	1.43	0.57	1.30	0.193
Psychoticism	1.41	0.39	1.29	0.42	8.83	<0.001

**Table 4 medicina-59-00971-t004:** Gender difference of subscales score on the SCL-90 of respondents.

Subscales	Female (*n* = 624)		Male (*n* = 401)		t	*p*
	M	SD	M	SD		
Somatization	1.63	0.66	1.61	0.69	0.465	0.642
Obsessive compulsive	1.91	0.68	1.81	0.69	2.285	0.023
Interpersonal sensitivity	1.59	0.61	1.56	0.65	0.749	0.454
Depression	1.69	0.66	1.60	0.68	2.105	0.035
Anxiety	1.55	0.60	1.56	0.66	−0.25	0.802
Hostility	1.62	0.62	1.55	0.65	1.484	0.138
Phobic anxiety	1.42	0.56	1.35	0.57	1.939	0.053
Paranoid ideation	1.45	0.53	1.46	0.61	−0.278	0.781
Psychoticism	1.39	0.50	1.42	0.59	−0.873	0.383
Sleeping and eating	1.64	0.63	1.65	0.67	−0.242	0.809
Total scores	144.22	50.32	140.77	55.02	1.032	0.302

## Data Availability

Not applicable.

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
