# Peer review of "Mental Health of Primary and Secondary School Teachers in the Remote Mountain Areas"

_medicina, 2023, doi:10.3390/medicina59050971_

Round 1

Reviewer 1 Report

The paper covers a relevant topic and is well-organized. The research design is adequate, and the results are presented clearly. As for suggestions, I would like to mention:

            It would be advantageous to include information about the magnitude of the observed differences and the statistical significance of the findings.

            The reference to ethics approval and consent to participate should follow the MDPI template form. Perhaps there is no need to include it in the methodology section. However, if it is included, the sequence of presentation should be reorganized.

Reviewer 2 Report

Mental health of primary and secondary school teachers in the remote mountain areas which is a topic of these paper (investigation) is a very important issue.  Most of research are based in urban areas, so this is an extremely appreciated

The study protocol was approved by the Chinese Clinical Trial Ethics Committee (NO. ChiECRCT2021009) and registered with the Chinese Clinical Trial Registry (NO. ChiCTR2100046396).  

Methodology and results are clearly writen and well presented. 

Limitations of this study are also desribed: the assessment instrument, and participants may have concealed their true psychological status for fear of private information leaks and the reputation of the school. In addition, the possible affected stressors were not surveyed. Also, this study was conducted in a remote

area of China which may have biased results.

Conclusions is short and clear. This study indicates that mental status of primary and secondary school teachers in

remote mountain area is not optimistic. Mental health

examinations might be incorporated into annual physical examinations to facilitate timely detection and early intervention of adverse emotions. 

I suggest to include some additional references and check english language. 

Reviewer 3 Report

This article is interesting and quite well done. The research investigates the mental health situation of teachers in a mountainous area of China, implementing a cross-sectional screening study.

Title: The title is succinct and avoids redundant phrases, so it is fine but perhaps the geographical area in which the research was made should be specified so that the communication is more precise.

Abstract/summary: I think you have to state the sampling strategy, sample size, response rate and main sample characteristics (age, sex breakdown, social class or other relevant properties) and you have to described the design of the study.

Introduction: You could say, even in brackets, the age of the children attending the primary and the secondary schools. I would like a little more explanation of the educational reform that has taken place, at what time it started and what the objectives and methods of the reform are and who it involves as subjects. Please, describe any necessary background information about the setting for the study and justifie the choice of measures and the sampling strategy.

Methods: Please, always use the past tense when describing other people's findings, and your own methods and results. Clarify well the meaning of primary / secondary / middle schools, through the age of the pupils or other concepts needed to understand the school system. Please, clearly liste the measures used, indicating where possible indices of validity and reliability or giving a citation where these can be found

Results: I’m not sure that the mentioned concept “baseline level” is correct since it is generally used in longitudinal studies to mean measurements at the beginning of the study.

Discussion: The discussion is very good and clear. It might be interesting to relate the results to characteristics of the schools considered or of the school system in general, e.g. the number of pupils per class, the hours per week and day, the age of retirement, the characteristics of the families there and the characteristics of the teachers' families, how much work in and out of the home burdens women.

Limitations of the study: address issues such as sample size, sample representativeness, measurement error, measurement bias. Perhaps another limitation could be the reference to the national average of the instrument used, which was calculated in the 1980s. Now it may have changed and the psychiatry textbooks have also changed. In any case, being an exploratory and screening study, the research is well done and brings to light some particular critical issues.

General: avoid colloquial expressions that would be confusing to an international readership and ensure that all abbreviations are spelled out in full the first time they are used.
